# Lung Cancer Screening in Greece: A Modelling Study to Estimate the Impact on Lung Cancer Life Years

**DOI:** 10.3390/cancers14225484

**Published:** 2022-11-08

**Authors:** Kyriakos Souliotis, Christina Golna, Pavlos Golnas, Ioannis-Anestis Markakis, Helena Linardou, Dimitra Sifaki-Pistolla, Evi Hatziandreou

**Affiliations:** 1Faculty of Social & Political Sciences, University of Peloponnese, 20100 Corinth, Greece; 2The Health Policy Institute, 15123 Maroussi, Greece; 34th Oncology Department & Comprehensive Clinical Trials Center Metropolitan Hospital, 18547 Athens, Greece; 4Fairlife LCC, 16675 Athens, Greece

**Keywords:** lung cancer, lung cancer life years lost, screened and linked to care, screening, lung cancer screening, modelling, low-dose computed tomography, tomography

## Abstract

**Simple Summary:**

Increasing screened and linked to care (SLTC) to a hypothetical 100% of eligible high-risk people in Greece leads to a statistically significant reduction in deaths and in total years lost due to lung cancer, when compared with the current SLTC paradigm. Over 5 years, the model predicted a difference of 339 deaths and 944 lost years between the hypothetical and current scenario. More specifically, the hypothetical scenario led to fewer deaths (−24.56%, *p* < 0.001) and fewer life years lost (−31.01%, *p* < 0.001). It also led to a shift to lower-stage cancers at the time of diagnosis. This study suggests that applying a 100% screening strategy amongst high-risk adults aged 50–80 would result in additional averted deaths and lung cancer life years lost (LCLYs) gained over 5-years in Greece.

**Abstract:**

(1) Background: Lung cancer causes a substantial epidemiological burden in Greece. Yet, no formal national lung cancer screening program has been introduced to date. This study modeled the impact on lung cancer life years (LCLY) of a hypothetical scenario of comprehensive screening for lung cancer with low-dose computed tomography (LDCT) of the high-risk population in Greece, as defined by the US Preventive Services Taskforce, would be screened and linked to care (SLTC) for lung cancer versus the current scenario of background (opportunistic) screening only; (2) Methods: A stochastic model was built to monitor a hypothetical cohort of 100,000 high-risk men and women as they transitioned between health states (without cancer, with cancer, alive, dead) over 5 years. Transition probabilities were based on clinical expert opinion. Cancer cases, cancer-related deaths, and LCLYs lost were modeled in current and hypothetical scenarios. The difference in outcomes between the two scenarios was calculated. 150 iterations of simulation scenarios were conducted for 100,000 persons; (3) Results: Increasing SLTC to a hypothetical 100% of eligible high-risk people in Greece leads to a statistically significant reduction in deaths and in total years lost due to lung cancer, when compared with the current SLTC paradigm. Over 5 years, the model predicted a difference of 339 deaths and 944 lost years between the hypothetical and current scenario. More specifically, the hypothetical scenario led to fewer deaths (−24.56%, *p* < 0.001) and fewer life years lost (−31.01%, *p* < 0.001). It also led to a shift to lower-stage cancers at the time of diagnosis; (4) Conclusions: Our study suggests that applying a 100% screening strategy amongst high-risk adults aged 50–80, would result in additional averted deaths and LCLYs gained over 5 years in Greece.

## 1. Introduction

Lung cancer is the first cause of cancer mortality, accounting for almost 1 in 5 deaths from cancer globally in both sexes, and responsible for ~270,000 deaths annually in Europe [1]. Most patients are diagnosed with advanced disease (>70%), while lung cancer is curable when diagnosed at initial stages and highly treatable in non-metastatic stages [2]. Lung cancer is etiologically associated with smoking and is a disease with a significant economic burden in societies [3].

In Greece, data from Globocan 2020, show that 8960 new lung cancer cases are diagnosed each year. Lung cancer deaths are the first cause of cancer-related deaths in the country, representing 23.1% of the total, with 7662 estimated deaths for 2020 [4]. Greece is also the second country in the European region, in tobacco smoking among adults >15 years old (Age-Standardized Prevalence: 34.3%) and cigarette smoking (Age-Standardized Prevalence: 38%) [5].

Recent large, randomised trials on low-dose computed tomography (LDCT) screening, including the American National Cancer Institute-sponsored National Lung Screening Trial (NLST) as well as the Dutch/Belgian NELSON trial, have shown significant reductions (20–26%) in lung cancer (LC) mortality and have triggered international efforts to implement LC screening [6,7,8]. European mortality data has recently become available from the NELSON randomised controlled trial, which confirmed lung cancer mortality reductions by 26% in men and 39–61% in women. A recently presented UK meta-analysis [9] of 4055 participants, receiving either a single invitation to LDCT screening or no screening (usual care) between October 2011 and February 2013, confirmed that LDCT screening reduces mortality from lung cancer. These results have mobilized advocacy for LDCT-LCS implementation in Europe. However, implementation of a lung cancer screening program is challenging and depends on many country-specific factors [10], including the assessment of the economic impact of LDCT-LCS and the definition of relevant guidelines. There is already evidence that LDCT screening has both clinical and cost-effectiveness benefits [8], although there is still some degree of uncertainty. For instance, Snowsill et al., highlight that a single round of LDCT-LCS may be cost-effective at conventional thresholds, however, there is still major uncertainty about the effect on costs and the magnitude of benefits [8]. These benefits and uncertainties are largely dependent on access to and availability of such services, particularly across countries with wide differences in geographic allocation of healthcare resources, as well as their actual useability and outcomes, to be monitored and reported upon with real-world evidence. Therefore, additional studies are required to determine the exact economic impact of LDCT screening for LC, by exploiting existing large-scale randomized clinical trials or other observational, population-based, or registry studies.

Given the high epidemiological burden of lung cancer in Greece, the current study aims to model the impact on lung cancer life years (LCLY) of a hypothetical scenario of comprehensive screening for lung cancer with LDCT of all high-risk population in Greece, as defined according to the US Preventive Services Taskforce protocol [11].

## 2. Materials and Methods

### 2.1. Verification Study

The program is implemented in a pipeline of several tailor-made functions. Each function was tested independently for the validity of its output using arbitrary data as input. Furthermore, several methods of static analysis were used for the whole program, such as control flow analysis and data flow analysis. Moreover, prior to final modeling, an estimation of execution times together with the usage of system resources was made, for project management purposes.

### 2.2. Modelling Study Objectives 

The modeling study objective was to estimate the impact on LCLY of adopting a hypothetical scenario, where 100% of the high-risk population in Greece, as defined by the US Preventive Services Taskforce [11], i.e., aged 50–80, first-hand smokers (20 pack-years) or ex-smokers (have quit within the past 15 years) would be screened and linked to care (SLTC) for lung cancer versus the current scenario of background (opportunistic) screening only (Figure 1).

### 2.3. Modelling Study Methods

We reviewed available literature on modeling lung cancer screening versus no screening [6,11,12,13,14]. We identified critical data to include in our model. We then developed a cohort stochastic model to monitor a hypothetical cohort of 100,000 high-risk men and women aged 50–80 first-hand smokers (20 pack-years) or ex-smokers (have quit within the past 15 years) over a horizon of 5 years in 3-month time steps (Figure 2).

Due to the lack of published data in Greece, we developed a de novo methodology to define probabilities of transitioning from one health state to another (without cancer, with cancer, alive, dead) based on input from clinical experts. Expected outcomes (cancer cases, deaths, lost LYs) were modeled both for the current and the hypothetical scenario, and the difference in lung cancer life years lost/gained between the two scenarios was calculated.

The impact of screening was assessed on lung cancer deaths only—therefore, all other lung cancer-unrelated causes of death were not considered in the model, thus avoiding skewing any model results.

Further, the model assumed no progression for cancer type Non-Small Cell Lung Cancer (NSCLC)/Stage 0 over the model time frame, as per clinical expert opinion. Screening after symptoms and random cancer-unrelated screenings were assumed to be performed in annual loops. This may underestimate model calculated benefits.

### 2.4. Modelling Study Data Inputs

We used data from Globocan, 2020 [5] and Hellenic Statistical Authority [15] on a smoking population to define the annual number of lung cancers to be identified amongst our high-risk cohort (incidence-based approach). This approach addresses the lack of data on the number of lung cancer screenings/false screenings, as cancer cases identified in the study cohort are the “actual” cases that would be incident amongst the cohort according to the published national lung cancer incidence. The clinical expert opinion provided data on clinical management, irrespective of sex or age.

### 2.5. Modelling Study Schema

In the current scenario, each year of the model (which comprises four quarters) the model controlled whether a person in the cohort was a lung cancer case (defined through lung cancer incidence), had developed symptoms, or had undergone a random cancer-unrelated scan. If the person had lung cancer (defined as “patient” for reference) and either developed symptoms or underwent a random CT scan, the model assumed that the patient was diagnosed and received optimal treatment & care, according to treatment guidelines. If the person had lung cancer (defined as “patient” for reference) but had not developed symptoms nor had he/she undergone a random CT scan, the model assumed that the patient was undiagnosed and continued in time as untreated until the next year. When lung cancer in this undiagnosed and untreated patient progressed, the patient had a defined probability to develop symptoms. Outside this probability, the patient remained undiagnosed and untreated until the next lung cancer stage or the next year (whichever occurred first), when the same process loop was applied. In the new scenario, all cohort members underwent LDCT screening upon entering the model and, if diagnosed with lung cancer, were identified as patients and assigned to optimal treatment and care, according to the latest available international, European, and national treatment guidelines [16,17,18]. The model followed the patient cohort for the next 5 years to estimate life years gained amongst those diagnosed as patients when this 100% SLTC approach was applied. The remainder of the cohort members, who were not diagnosed at point 0, were screened (100%) for lung cancer again on an annual basis. The same process loop as above was applied. The model schema –cohort population flow is depicted in Figure 3

### 2.6. Model Transitions’ Structure

In the current treatment scenario, each person entering the model was assigned a health state, either healthy or cancer patient, based on the health state distribution presented in Appendix A. Following this assignment, healthy cohort members moved forward to the next model year. At that moment, their health state was re-evaluated according to the procedure described above, and if they remained healthy, this process continued until model time ended when the person exited the model healthy.

However, if, at any moment, the health state of the person changed to being a cancer patient, then a different routine was applied. If the patient underwent a random cancer-unrelated CT scan or developed symptoms, he was assigned cancer patient status. The probability of random screening was set to 3.84% for the Greek population over 15 years old [1,2,3], as calculated from data shown in Appendix A, and the probability of developing symptoms, by cancer type and stage, was set according to Appendix A. If the patient was diagnosed with cancer, he/she was assumed to receive optimal treatment and was denoted as a “treated patient”. Otherwise, the patient would be undiagnosed and denoted as an “untreated patient”.

For “treated patients”, the progression of cancer depended on their type and stage. This progression was quantified by assuming that the patient moved from the initial stage to another pseudo-stage. The NSCLC-Stage 0 patients remained at this stage until they exited the model. The remaining patients moved to the next stage at a certain time according to Appendix A. In addition, these patients were evaluated every three months by the model to assert where they remained alive and at the same pseudo-stage, had died or moved to another pseudo-stage, earlier than expected, according to probabilities shown in Appendix A. This process continued, even if the patient moved to another stage, until the patient died or the model time ended, whichever occurred first.

As regards “untreated patients”, transitions from stage to stage remained as above and presented in Appendix A, depicting a faster cancer progression or death, as these cohort members received no treatment and care. In the hypothetical scenario, where all cohort members underwent LDCT screening upon entering the model, health states were distributed according to Appendix A. Cancer was diagnosed at the same time cancer occurred, and patients received optimal treatment and care (Appendix A, treated patients).

The programming routines, which implement the numerical model, were developed in MatLab (MathWorks Inc., Apple Hill Drive, MA, USA) computational platform. A total of 150 iterations of simulation scenarios were conducted for 100,000 persons each. Any treatments that commence the following screening continue until death or model time stop, whichever occurs first.

## 3. Results

The modeling study revealed that increasing SLTC to a hypothetical 100% of eligible high-risk people in Greece leads to a statistically significant reduction in deaths and in total years lost due to lung cancer, when compared with the current SLTC paradigm. Over 5 years, the model predicted a difference of 339 deaths and 944 lost years between the hypothetical and current scenario (Table 1 and Table 2). More specifically, the hypothetical scenario led to fewer deaths (−24.56%) and fewer life years lost (−31.01%) (Table 3). It also led to a shift to lower-stage cancers at the time of diagnosis, allowing for more frequent eligibility for curative treatment.

## 4. Discussion

This modeling study estimated the clinical impact in terms of deaths averted and life years gained from applying a hypothetical 100% screening and linkage to care scenario on a cohort of LC high-risk persons, as defined by the US Preventive Services Task Force, in Greece. It analyzed two screening and disease management scenarios: a current scenario, where a high-risk person may undergo a random cancer-unrelated screening or experience symptoms after being assigned the probability to be a lung cancer patient, and a completely hypothetical scenario, in which all high-risk persons are screened for lung cancer upon entry in the model. In both scenarios, post-screening, patients are linked to treatment until death or model time end, whichever occurs first. The new scenario led to improved outcomes, in terms of fewer deaths (−24.61%) and fewer life years lost (−30.90%) over 5 years. The model also indicated that applying the hypothetical scenario leads to a substantial shift to lower-stage cancers at the time of diagnosis, therefore, more patients would be eligible for curative treatment (mainly surgical).

Despite the study using a stochastic model based on clinical expert opinion, findings are in line with the recently published outcomes of the NELSON clinical trial [6], which showed a reduction in lung cancer-related deaths amongst men aged 50–74 of −24% (95% CI, −39% to −6%) and amongst women aged 50–74 of −33% (−62% to +14%), when these patients were screened at time T0, year 1, year 3 and year 5.5.

As promising as these data may appear in reducing mortality, there are still open questions about introducing universal screening for high-risk people, mainly regarding cost-effectiveness and overdiagnosis. Furthermore, experience from around the globe revealed a range of barriers, including primarily low rates of acceptance and uptake of screening from referring primary physicians, and the general population as well as many difficulties in standardizing techniques and processes among centers and radiologists [19,20,21,22,23]. These indicate that there may be a need to introduce formal, national screening programs, if to achieve higher uptake.

Although many guidelines have been issued in Europe and guidance has been published to promote and assist with the implementation of LCS programs, only Croatia [24], Poland [25], and the Czech Republic [26] have, to date, initiated such national programs, while others have set up pilot schemes in areas with high smoking prevalence [24]. The UK Lung Screening Trial (UKLS) [27] used population primary care records to identify those aged 50–75 years and asked them to respond to a questionnaire. Only 11.5% of respondents were at a high enough risk to be considered eligible for the trial, thus underlining the reluctance of high-risk groups to participate in such initiatives [28]. In Manchester [29], a pilot of a community-based lung cancer low-dose CT screening concluded that the program made a cost-effective use of limited National Health System (NHS) resources. Still, the authors highlighted that more research is required on the exploration of the conditions such a program could enhance patient health while remaining cost-effective.

In addition, LDCT screening of eligible people should incorporate sufficiently described and appropriately integrated linkage to care services for evaluation of screening outcomes, i.e., suspect pulmonary nodules and linkage to care as required, after assessment by a multidisciplinary team. Such Pulmonary Nodule Clinics (PNCs) [30,31] have been established in the US [30] and the UK [31] and are being shown as highly effective in optimizing lung cancer screening public health potential. A single healthcare system retrospective review by Melton et al. [32], which compared the staging of cancers detected through a PNC and the rest of the general lung cancer population in a community healthcare system in Central Pennsylvania revealed significantly earlier stages of lung cancer at diagnosis in the PNC, coupled with higher five-year survival rates for patients so diagnosed.

Nonetheless, such clinics are still underdeveloped or only informally functioning, mostly unconnected to national health systems, in several countries, including Greece. Any national LCS initiative must be coupled with adequately defined, appropriately aligned, and standardized assessment and linkage to care services, as may be provided by PNCs, in physical or virtual form, to ensure there is optimal follow-through of any identified cases as well as sufficient consultation support in cases where a follow-up monitoring schedule should be established. This will be highly dependent on the specificities of each health system and should be taken into consideration early in the design of any LCS program. Further, LDCT screening offers new opportunities for radiologists collaborating with any such PNCs for the early detection of other diseases [33], such as cardiovascular disease and chronic obstructive pulmonary disease, which are even more frequent causes of death than lung cancer [33].

### Limitations

This study is a modeling study based on a stochastic model that was informed by clinical expert opinion. Lung cancer incidence and smoking prevalence were based on published literature and official sources. The model does not include background mortality, so as not to skew model results—the model produces results only for lung cancer-related deaths. The clinical expert opinion provided data on clinical management questions irrespective of sex/age. Therefore, all cohort members are treated as the same, not allowing for inter-sex or age variations. Additionally, we assumed no progression for cancer type NSCLC/Stage 0 over the model time frame, as per clinical expert opinion. This may impact model results. Finally, screening after symptoms and random cancer-unrelated screenings were assumed to be performed in annual loops and this may underestimate model calculated benefits.

Future work could manage the aforementioned limitations by collecting real-world evidence on the distribution of lung cancer cases in terms of sex and age, and the current screening behaviors and practices.

## 5. Conclusions

Lung cancer causes a substantial epidemiological burden in Greece. Yet, no formal national lung cancer screening program has been introduced to date. Our study suggests that applying a 100% lung cancer screening strategy with LDCT amongst high-risk adults aged 50–80, would result in additional averted deaths and LCLYs gained over 5 years in Greece. Such screening should, however, be coupled with the introduction of integrated assessment and linkage to care services, to ensure optimal achievement of expected public health outcomes.

## Figures and Tables

**Figure 1 cancers-14-05484-f001:**
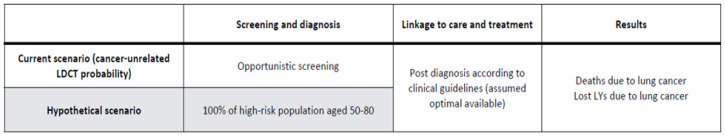
Current and hypothetical scenario for LCLY.

**Figure 2 cancers-14-05484-f002:**
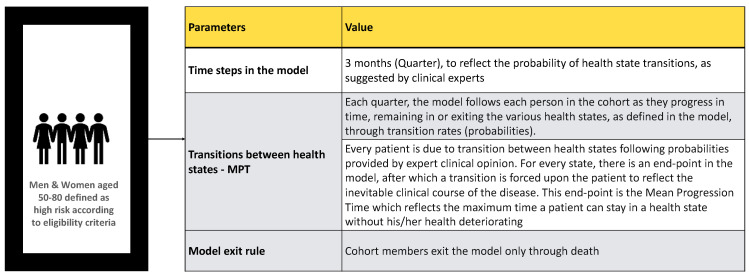
Design of the hypothetical model inputs.

**Figure 3 cancers-14-05484-f003:**
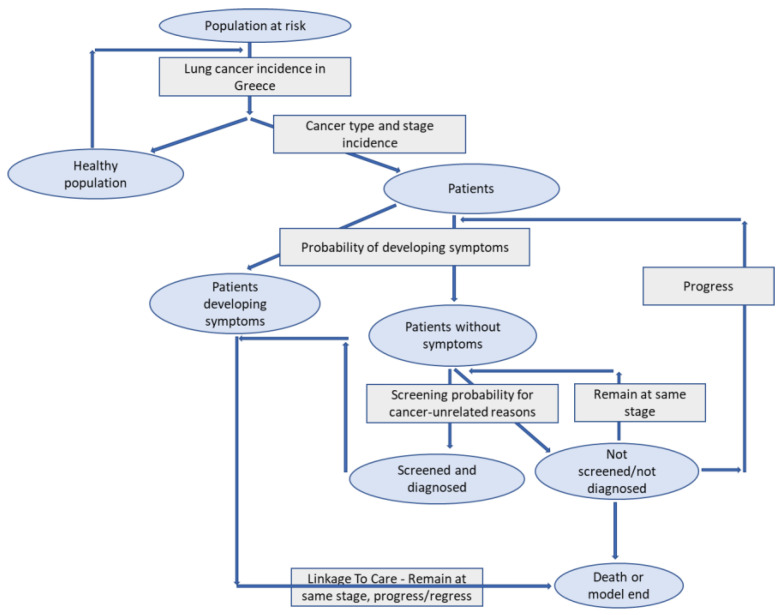
Model schema for the cohort population flow.

**Table 1 cancers-14-05484-t001:** Current scenario modeling results.

Parameter	Mean Value	Standard Deviation
Cancer cases (per model life cycle)	1929.07	39.57
Cancer-related deaths (per model life cycle)	1371.21	34.68
Lost years (per model life cycle)	3042.84	90.98

**Table 2 cancers-14-05484-t002:** Hypothetical scenario modeling results.

Parameter	Mean Value	Standard Deviation
Cancer cases (per model life cycle)	1925.55	41.00
Cancer-related deaths (per model life cycle)	1031.89	31.54
Lost years (per model life cycle)	2098.85	77.37

**Table 3 cancers-14-05484-t003:** Comparison between scenarios.

Parameter	Difference (%)	*p*-Value
Cancer-related deaths (per model life cycle)	−24.61%	<0.001
Lost years (per model life cycle)	−30.90%	<0.001

## Data Availability

Primary data are not available due to data privacy purposes.

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
