# Peer review of "Lung Cancer Screening in Greece: A Modelling Study to Estimate the Impact on Lung Cancer Life Years"

_cancers, 2022, doi:10.3390/cancers14225484_

Round 1

Reviewer 1 Report

This is a well-written, well-structured paper which examines a hypothetical scenario of comprehensive screening for lung cancer with LDCT in Greece.

Some Suggestions:

1) Please be more precise of what statistical analyses did you use for the specific paper. The Phrase “several methods of static analysis were used to the whole program” in 2.1 seems a bit generic. Please also mention any tests that you have performed and did not produce significant results.

2) For 2.2, 2.3 and 2.4, it is welcome to add a couple of references from previous studies which might used the same or similar methodology (approach) with yours.

3) In Figure 3, it is preferable to have a one “end state”. As I read it I found 2 terminate states the “Remain at same stage, progress/regress until death or model end” and the “death or model end” you may consider to merge or add an extra arrow between them in order to have one final state.

4) Line 174 please erase the double space.

5) For tables 1-3, it is not clear on me what the mean of the “Cancer cases” presents. Is the Cancer cases per year? Per 3-months? Please be more precise in the tables’ labels.

6) Future work (especially related to the proposed model) is also welcome to be presented after the study's limitations.

Author Response

This is a well-written, well-structured paper which examines a hypothetical scenario of comprehensive screening for lung cancer with LDCT in Greece.

 Some Suggestions:

1) Please be more precise of what statistical analyses did you use for the specific paper. The Phrase “several methods of static analysis were used to the whole program” in 2.1 seems a bit generic. Please also mention any tests that you have performed and did not produce significant results.

Re:  Thank you for your comment. In this section of the manuscript, we present the verification method we used to validate the software code, prior to running the model. The static analysis includes processes such as control flow and data flow analysis. Therefore, this section does not refer to the statistical analysis mentioned elsewhere in the manuscript.

2) For 2.2, 2.3 and 2.4, it is welcome to add a couple of references from previous studies which might used the same or similar methodology (approach) with yours.

Re: Thank you for your comment. We have added references and amended this section of the manuscript in line with reviewer recommendations.

 3) In Figure 3, it is preferable to have a one “end state”. As I read it I found 2 terminate states the “Remain at same stage, progress/regress until death or model end” and the “death or model end” you may consider to merge or add an extra arrow between them in order to have one final state.

Re: We have redesigned Figure 3 to reflect reviewer’s comment.

4) Line 174 please erase the double space.

Re: Thank you for noticing. We have erased it.

5) For tables 1-3, it is not clear on me what the mean of the “Cancer cases” presents. Is the Cancer cases per year? Per 3-months? Please be more precise in the tables’ labels.

Re: Thank you for your comment. We have now added the wording “(model life cycle)” to clarify the time-period used.

 6) Future work (especially related to the proposed model) is also welcome to be presented after the study's limitations.

Re: Thank you for your comment. We have now added our thoughts on future work, after the study’s limitations, as per reviewer’s suggestion.

Reviewer 2 Report

Dear Authors, many thanks for the opportunity to read your valuable paper.

The introduction is well done, the methods are well exposed, and the results are excellent.

Discussion is approriate.

The entire paper is well written, and it deserves to be published.

I think it is a good job.

I have only to suggest to you to include the supplementary table 4 as main table or figure in the result section, that in my point of view is appropriate.

I have only to suggest to you to include this paper, that in my point of view is appropriate. (you are free to do, I'm not the author)

Best regards

Author Response

Dear Authors, many thanks for the opportunity to read your valuable paper. The introduction is well done, the methods are well exposed, and the results are excellent. Discussion is approriate. The entire paper is well written, and it deserves to be published. I think it is a good job. I have only to suggest to you to include the supplementary table 4 as main table or figure in the result section, that in my point of view is appropriate.

I have only to suggest to you to include this paper, that in my point of view is appropriate. (you are free to do, I'm not the author)

Best regards

Re: Thank you for your comments and your suggestion. We have initially considered adding Table 4 of the supplementary material in the main body of the manuscript, but unfortunately it would necessitate we move all supplementary tables to the main manuscript for consistency purposes, and we would then exceed the tables/figures limit in the authors’ guidelines.  

Reviewer 3 Report

The goal of the manuscript by Kyriakos Souliotis and colleagues was to

Emphasize the positive outcomes of low-dose computed tomography (LDCT) screening to determine the possibility of developing lung cancers, in the high-risk adult population (specifically) in Greece. For the same using the existing data for the current scenario and then generating a simulation model, the authors predict that LDCT of high-risk adults helps in reducing the deaths caused by lung cancer and helps to increase the number of lung cancer life years.

While these observations may be potentially useful, the manuscript has some weaknesses as follows:

1)    For the authors, it is recommended to look at the published article - Added benefits of early detection of other diseases on low-dose CT screening (PMID: 33718052). This article similarly mentions the benefits of LDCT screening for lung cancers, lung fibrosis, and other diseases.

The manuscript by Kyriakos Souliotis and colleagues seems a repetition of similar work, but more focused on the adults from a particular region (example in this manuscript – Greece).

The authors also fail to use it as a reference.

2)    The authors do mention very briefly, the economic impact of these studies. More information or correlation on how it can affect health services available in remote areas, insurance policies and providers, etc. will add substantial support to the study.

3) The author needs to provide some additional information on the demographics of the adult population to be taken into consideration, with respect to the treatments the adult is going through

Author Response

The goal of the manuscript by Kyriakos Souliotis and colleagues was to emphasize the positive outcomes of low-dose computed tomography (LDCT) screening to determine the possibility of developing lung cancers, in the high-risk adult population (specifically) in Greece. For the same using the existing data for the current scenario and then generating a simulation model, the authors predict that LDCT of high-risk adults helps in reducing the deaths caused by lung cancer and helps to increase the number of lung cancer life years.

While these observations may be potentially useful, the manuscript has some weaknesses as follows:

Re: Thank you for the review and your comments and recommendations.

1)For the authors, it is recommended to look at the published article - Added benefits of early detection of other diseases on low-dose CT screening (PMID: 33718052). This article similarly mentions the benefits of LDCT screening for lung cancers, lung fibrosis, and other diseases. The manuscript by Kyriakos Souliotis and colleagues seems a repetition of similar work, but more focused on the adults from a particular region (example in this manuscript – Greece). The authors also fail to use it as a reference.

Re: We have added the reference and relevant text in the discussion section of the manuscript, as per reviewer’s comments.

2)    The authors do mention very briefly, the economic impact of these studies. More information or correlation on how it can affect health services available in remote areas, insurance policies and providers, etc. will add substantial support to the study.

Re: Thank you for the comment. We have amended the text in our introduction, to address reviewer’s comment.

3) The author needs to provide some additional information on the demographics of the adult population to be taken into consideration, with respect to the treatments the adult is going through

Re: As mentioned in the methodology section of our study, this is a modelling study that follows a hypothetical cohort of high-risk adults aged 50-80 as they progress through either screening or no screening for lung cancer over 5 years. Actual data on distribution of such a cohort according to age and sex are not available. Further, in Greece, treatment for lung cancer screening follows the most recent NCCN and ESMO guidelines, as included in the EOPE local therapeutic protocols, and there is wide access to innovative cancer therapies, through a system of electronic approval of reimbursement for such therapies. Therefore, our model assumes linkage to optimal care and treatment, post diagnosis, according to specific patient characteristics. We have added a clarification in the text in line with reviewer’s comments.

Round 2

Reviewer 3 Report

Hello Kyriakos Souliotis and colleagues,

Thank you for accepting the review comments and adding the information to the manuscript.

Best wishes.

Reviewer.